analytical chemistry/computational chemistry/ green chemistry

hybrid micelle liquid chromatography, log *P* values, lipophilicity, aripiprazole, pramipexole, piribedil

**Author for correspondence:**
M. E. K. Wahba
e-mail: marywahba5@gmail.com

# Studying the suitability of hybrid micelle liquid chromatography for estimating the lipophilicity of some partial dopamine agonists used to attain the reward circuit

M. E. K. Wahba[1,2], D. El Wasseef[1,3] and D. El Sherbiny[1,3]

[1]Department of Pharmaceutical Chemistry, Faculty of Pharmacy, Delta University for Science and Technology, 35712 Gamasa, Egypt
[2]Department of Pharmaceutical Analytical Chemistry, and [3]Department of Medicinal Chemistry, Faculty of Pharmacy, Mansoura University, Mansoura 35516, Egypt

MEKW, 0000-0002-1352-0297; DES, 0000-0002-9829-3563

Three micellar-based mobile phases were developed and optimized for the simultaneous determination of certain partial dopamine agonists that are used to overcome the withdrawal symptoms of abused drugs, namely aripiprazole, pramipexole and piribedil. The studied drugs were separated using micellar liquid chromatography, hybrid micellar liquid chromatography (HMLC) and microemulsion liquid chromatography (MELC). The three developed mobile phases were studied to estimate their suitability for the measurement of log *p*-values of the studied drugs. Experimental determination of log $P_{m/w}$ values using the three mobile phases demonstrates that HMLC is the mobile phase of choice since the obtained practical log $P_{m/w}$ values were in accordance with the reported log *P* values, and calculated log *P* and log *D* values. An explanation of the obtained results was presented based on the separation retention mechanism for each chromatographic technique. Furthermore, the effect of the pH and the column temperature in HMLC on the practical log $P_{m/w}$ values was studied. To verify its suitability for experimental measurement of log $P_{m/w}$, HMLC was subjected to full validation according to the United States Pharmacopeia.

# 1. Introduction

Amphetamine, methamphetamine and cocaine are ranked by the world drug report [1] as the most prevalent psycho-stimulant illicit abused drugs. Their CNS-stimulating effect is produced by increasing the synaptic concentration of neurotransmitters like dopamine, noradrenaline and serotonin [2]. The reward circuit, a terminology referring to the enhancement of dopamine secretion in the meso-limbic portion of the brain, is intimately connected to behavioural reinforcement [2–5], while this effect is not induced by elevation in the concentration of other neurotransmitters; noradrenaline and serotonin [6–8]. The idea of using partial dopamine agonists (PDA) as therapeutic agents to overcome the withdrawal symptoms of drug abuse has been supported by examination in drug addiction animal models [9]. The rationale for this unique therapeutic effect arises from the fact that PDA possesses high affinity but low inherent activity for dopamine receptors. Consequently, this pharmacological action of PDA mainly depends on the existing dopamine at the synapses [9]. A significant impact of structure–activity relationship (SAR), based mainly on chemical structure and physico-chemical properties of PDA, imparts this unique pharmacological behaviour to such pharmaceutical compounds [8–10].

It is well established that bioavailability and specific binding of drugs to the target organs are significantly related to their lipophilicity [11–14]. A traditional method for the estimation of this vital parameter is the one introduced by Hansch *et al.* [15,16], where the researchers measured partitioning of the concerned pharmaceutical agent between 1-octanol and water, and assumed its logarithm (log $P_{o/w}$) to be a numerical value reflecting its lipophilic nature. In spite of its popularity, this technique suffered from many drawbacks [11]. The pronounced physico-chemical variations between 1-octanol/water system and *in vivo* biological system is considered one of the major limitations of the resultant log $P_{o/w}$ values [11]; hence, alternative analytical tools were needed for its relevant determination. Liquid chromatography has been introduced as an effective analysis aid for this intended purpose [11]; in spite of this, when liquid chromatography was applied using the reversed-phase mode—where the mobile phase was prepared as a purely aqueous solution—imprudent retention of analytes was encountered, rendering the calculation of log $P_{o/w}$ values for some pharmaceuticals non-viable [11]. Alternatively, micelle-based chromatographic techniques have replaced traditional RP-HPLC to overcome this obstacle [11,12,17–20]. The mobile phases used in these techniques are basically composed of aqueous solutions containing micelles over their critical micelle concentrations, where the studied analytes partition between the aqueous phase and the micelles [11,12], where a well-established correlation between the resultant log $P_{m/w}$ and log $P_{o/w}$ values has been documented [21]. This work is concerned with the calculation of log $P_{m/w}$ values for some PDA drugs since this physico-chemical parameter affects their unique pharmacological action as previously mentioned. Three PDA were taken as model examples, including aripiprazole (ARP) 7-[4-[4-(2,3-dichlorophenyl)piperazin-1-yl]butoxy]-3,4-dihydro-1H-quinolin-2-one [22] (table 1), pramipexole (PRM) [(RS)-4,5,6,7-tetrahydro-N-6-propyl-2,6-benzothiazolediamine dihydrochloride monohydrate] [22] (table 1) and piribedil (PRB) 2-[4-(1,3-benzodioxol-5-ylmethyl) piperazin-1-yl] pyrimidine [22] (table 1).

Three micelle-based mobile phases were used during this study, namely MLC, HMLC, which involves the introduction of small ratios of organic modifiers to MLC, and MELC, which entails the addition of oil to an aqueous solution stabilized by a surfactant (mainly sodium dodecyl sulfate, SDS), and a co-surfactant, eventually producing an isotropically clear solution.

The hypothesis applied for experimental deduction of log $P_{m/w}$ is under the guidance of a previously published article concerned with a similar approach [12], where the critical micelle concentration (CMC), i.e. the concentration above which micelles are formed, of SDS at different temperature settings was obtained from a previously published report [24].

Comparison of the obtained log $P_{m/w}$ values of the investigated analytes with the reported and the calculated log $P$ values was carried out, proving that HMLC is the most suitable assay tool for log $P_{m/w}$ calculation among other micelle-based chromatographic techniques. Based on these findings, different HMLC separation conditions that may influence the experimentally determined log $P_{m/w}$ values were investigated, including the pH of the mobile phase, and the column temperature.

# 2. Experimental procedure

## 2.1. Instrumentation

All separations were carried out on a Shimadzu SPD-20A, with a loop of 20 µl capacity, and a UV-VIS detector (model SPD-20A) was used to carry out separation. The instrument is provided with a

**Table 1.** Physico-chemical properties of the studied drugs.

| drug | chemical structure | $pk_a$ [23] |
|------|--------------------|-------------|
| ARP | | 10 |
| PRM | | 10.5 |
| PRB | | 6.95 |

column oven (CTO-20A) and degasser unit (DGU-207). Serial no.: L 20135330505 AE, 220–230/240 V, 50–60 HZ, 160 VA, Kyoto, Japan. pH meter, Jenway, UK.

Log $P$ and log $D$ values were calculated using Advanced Chemistry Development (ACD/Labs) Software v. 11.02 (© 1994–2020 ACD/Labs), accessed on 15 October 2020.

## 2.2. Materials and reagents

Aripiprazole (ARP), Sigma Aldrich, USA. Pramipexole (PRM), Eva Pharm, for pharmaceuticals and medical appliances, S.A.E., Giza, Egypt. Piribedil (PRB), Servier Egypt industries limited, 6 October city, Giza, Egypt, under licence of Les Laboratories, Servier-France.

SDS, tri-ethyl amine (TEA), 1-octane, and 1-propanol (HPLC grade), Sigma Aldrich, USA.

Acetonitrile and methanol (HPLC grade), TEDIA, Fairfield, USA.

Ortho-phosphoric acid (min. assay 85%) was purchased from El Nasr Pharmaceutical Chemical Company, Egypt.

## 2.3. Chromatographic separation conditions

Routine steps for system stability were followed before and after analysis. Ultrafiltration of the mobile phases was carried out using nylon membrane filters of 0.45 µm pore size (Chrom Tech UK), followed by sonication for at least 30 min. After analysis, washing with ultra-filtered double distilled water for 30 min was performed, followed by washing with a mixture of methanol : double distilled water (1 : 1, v/v), for the next 30 min.

Three micelle-based mobile phases were developed for the separation of this mixture. In all three cases, the separation was carried out on a Prontosil Kromaplus C18 column (250 × 4.6 mm) of particle size 5 µm, Leonberg, Germany, which was kept at 40°C. Micellar mobile phase consisted of 0.18 M SDS containing 0.3% TEA and adjusted with 2 M ortho-H$_3$PO$_4$ to pH 6, the addition of 10% acetonitrile produced HMLC mobile phase, while the addition of both 1% octane and 10% acetonitrile resulted in MELC mobile phase. Separation of the three drugs was accomplished using 254 nm as a detection wavelength, adopting a flow rate of 1.5 ml min$^{-1}$.

## 2.4. Calibration curves construction

Quantitative analysis of ARP, PRM and PRB was established through the construction of a calibration curve for each analyte applying HMLC as an analysis technique. Stock solutions for both ARP and PRB were prepared as 1.0 mg ml$^{-1}$, while that of PRM was prepared as 500.0 µg ml$^{-1}$, using methanol as a solvent. Suitable volumes of the stock solutions were transferred to 10.0 ml volumetric flasks, followed by dilution to the mark with methanol, to prepare working solutions of the studied drugs

within linearity ranges of 1.0–200.0, 0.5–100.0 and 1.0–250.0 µg ml$^{-1}$ for ARP, PRM and PRB, respectively. Each working solution was analysed three times to take the average response. Calibration curves were then constructed by plotting the concentration of each drug (in µg ml$^{-1}$) versus peak area, followed by deduction of the regression equation of each drug. Laboratory-prepared mixtures for the three drugs in different concentrations were prepared by transferring the appropriate volumes of each stock solution into 10.0 ml volumetric flask, and completing the volume with methanol, taking the average peak area for each sample after triplicate injection. The concentration of each drug was then derived from its previously concluded regression equation.

## 2.5. Experimental determination of log $P_{m/w}$ of the studied drugs using MLC, HMLC and MELC

Different actual SDS concentrations for each mobile phase (after subtracting the CMC of SDS at 40°C) were plotted versus the inverse of the capacity factor for each drug, using the optimized chromatographic conditions for MLC, HMLC and MELC, where pH was adjusted to 6, and the column temperature was kept at 40°C for the three separation methods. Log $P_{m/w}$ values of ARP, PRM and PRB using the three mobile phases were calculated by considering it as the slope of the obtained regression equation for each drug.

## 2.6. Experimental determination of log $P_{m/w}$ of the studied drugs at different pH values using HMLC

Five sets of HMLC mobile phases were prepared; each set has a constant SDS concentration (either 0.05, 0.075, 0.1, 0.15 or 0.18 M), and each set is prepared with different pH values (ranging from 3 to 6). Actual concentrations of SDS in the five sets (X-axis) were plotted versus the inverse of the capacity factor for each of ARP, PRM and PRB (Y-axis), where analysis was performed keeping the column temperature at 40°C. These steps allowed predicting the variation of experimentally determined log $P_{m/w}$ with pH. Moreover, the correlation between the resultant log $P_{m/w}$ and the calculated dissociation constant (log $D$) for each pharmaceutical at each individual pH could be introduced.

## 2.7. Experimental determination of log $P_{m/w}$ of the studied drugs at different temperature settings using HMLC

The same procedures mentioned under §2.6 were followed to determine log $P_{m/w}$ for each drug at each temperature setting (25, 30, 35 and 40°C), where the pH of the mobile phase was kept at 6. This step allowed relating partitioning of the concerned drugs between the aqueous phase and micelles at different temperature settings.

# 3. Results and discussion

## 3.1. Separation parameters optimization

A thorough study was carried out to achieve optimum separation parameters and highest sensitivity measures for the three analytes using the three concerned mobile phases.

All separations were carried out on a Prontosil Kromaplus C18 column operating at 40°C. This column belongs to the monomeric type where incorporation of the C18 alkyl chain to a single atom of silica on the silica gel backbone is the main route of manufacture, which results in a highly stable column (can stand pH from 1 to 10) with improved reproducibility. Moreover, this column is full end-capped, which reduces the number of residual silanol groups, inhibiting their ionic binding with basic analytes, and consequently limiting the tailing of peaks.

The effect of column temperature on the separation performance for the three mobile phases was studied. It was found that by increasing the applied temperature (25–40°C), a subsequent decrease in the retention times of all the drugs resulted. Consequently, 40°C was selected to shorten the analysis time. Furthermore, the applied temperature setting enhanced the repeatability of the obtained retention times $t_r$ and peak responses of the studied pharmaceutical compounds. Moreover, a flow rate of 1.5 ml min$^{-1}$ was used, aiming to enhance peaks symmetries and attain shorter chromatographic runs.

To obtain the highest sensitivity measures, different wavelength settings were investigated (220–270 nm). To compromise between the different sensitivities of the three analytes, 254 nm was selected as an optimum detection wavelength.

As a conclusion, for the three used mobile phases, C18 column was thermostatically kept at 40°C, applying a flow rate of 1.5 ml min$^{-1}$, and adopting 254 nm as a detection wavelength.

Concentrations of SDS ranging from 0.05 to 0.18 M were investigated, 0.05 M succeeded to elute the three drugs but with long unacceptable retention times. By increasing the SDS concentration, the retention times of all three compounds decreased without affecting the peak responses, so 0.18 M SDS was chosen.

The mobile phase pH over the range of 3–6 was also studied. By elevating the pH, the $t_r$ of the three studied pharmaceuticals was slightly affected. This behaviour is concordant with their $pk_a$ values (table 1) [23]; being basic in nature, the drugs will not be affected by the change in pH as they are completely ionized over the studied pH range. Meanwhile, tailing of peaks at lower pH values was observed, which could be attributed to the interaction of the cationic drugs with the residual silanol groups on the stationary phase, keeping in mind that intense positive charges are expected to be carried by the analytes at lower pH values. By elevating the pH of the mobile phase through the addition of TEA, peaks symmetry was achieved, which could be ascribed to the reduction in the cationic charges carried by the studied pharmaceuticals, and consequently, their liability to the interaction with silanol groups is expected to decrease, in addition to the vital role of TEA in masking the residual silanol groups on the stationary phase. Accordingly, pH 6 was chosen for the separation of the investigated compounds as it resulted in symmetric peaks.

Studying these parameters permits simple preparation of MLC mobile phase from 0.18 M SDS containing 0.3% TEA and adjusted to pH 6 which succeeded to separate ARP, PRM and PRB at retention times of 6.4, 10.35 and 12.48 min, respectively (figure 1a).

Proceeding to prepare HMLC, an organic modifier should be included. Three organic modifiers were investigated: methanol, 1-propanol and acetonitrile. Methanol resulted in a remarkable delay in the elution of ARP and PRM, while PRB was retained on the column for more than 45 min. 1-propanol, on the other hand, succeeded to separate PRM and PRB but failed to separate ARP from the solvent front. Acetonitrile was, therefore, the organic modifier of choice, since it resulted in well-separated peaks in a reasonable analysis run time.

Acetonitrile ratio was also studied (from 7 to 15%). It was found that $t_r$ of the three compounds was slightly affected over the studied ratio. For environmental safety, 10% acetonitrile was used for this separation.

As a conclusion, the composition of the HMLC mobile phase to separate ARP, PRM and PRB is composed of 0.18 M SDS, 10% acetonitrile, containing 0.3% TEA, and adjusted to pH 6 with 2 M o-phosphoric acid. This mobile phase resulted in the separation of the three drugs at corresponding $t_r$ of 4.4, 7.6 and 11.4 min, respectively (figure 1b).

By including 1% octane to the previously prepared HMLC mobile phase, the MELC mobile phase was simply prepared, resulting in an enhanced reduction in the $t_r$ of all the drugs, i.e. 3.45, 5.6 and 6.8 min for ARP, PRM and PRB, respectively (figure 1c).

It is to be noted that in spite that the lipophilicity of ARP > PRM > PRB as reflected from their reported log P values [23], their elution order is reversed applying the three mobile phases. This retention attitude could be ascribed to mass transfer which is a unique phenomenon experienced by lipophilic drugs analysed by micelle-based mobile phases [25–27]. The main mechanism involved is the transport of highly lipophilic drugs from micelles to stationary phase, resulting in a remarkable reduction in their retention times. Keeping in mind that three studied pharmaceuticals are expected to be cationic through the entire analysis applying the three studied mobile phases, it is expected that their elution pattern is governed by their lipophilicities; and hence by mass transfer. Thus, the most lipophilic analyte, i.e. ARP, will be the most liable compound to mass transfer and elute first, followed by PRM, and eventually by PRB.

## 3.2. Studying the suitability of MLC, HMLC and MELC for experimental determination of log $P_{m/w}$

By referring to the plotted figures (figure 2a–c), and the obtained data in table 2, it is obvious that the experimentally determined log $P_{m/w}$ values for the three studied analytes applying HMLC (4.729, 2.323 and 1.648 for ARP, PRM and PRB, respectively) are the closest to the reported log P values (4.6,

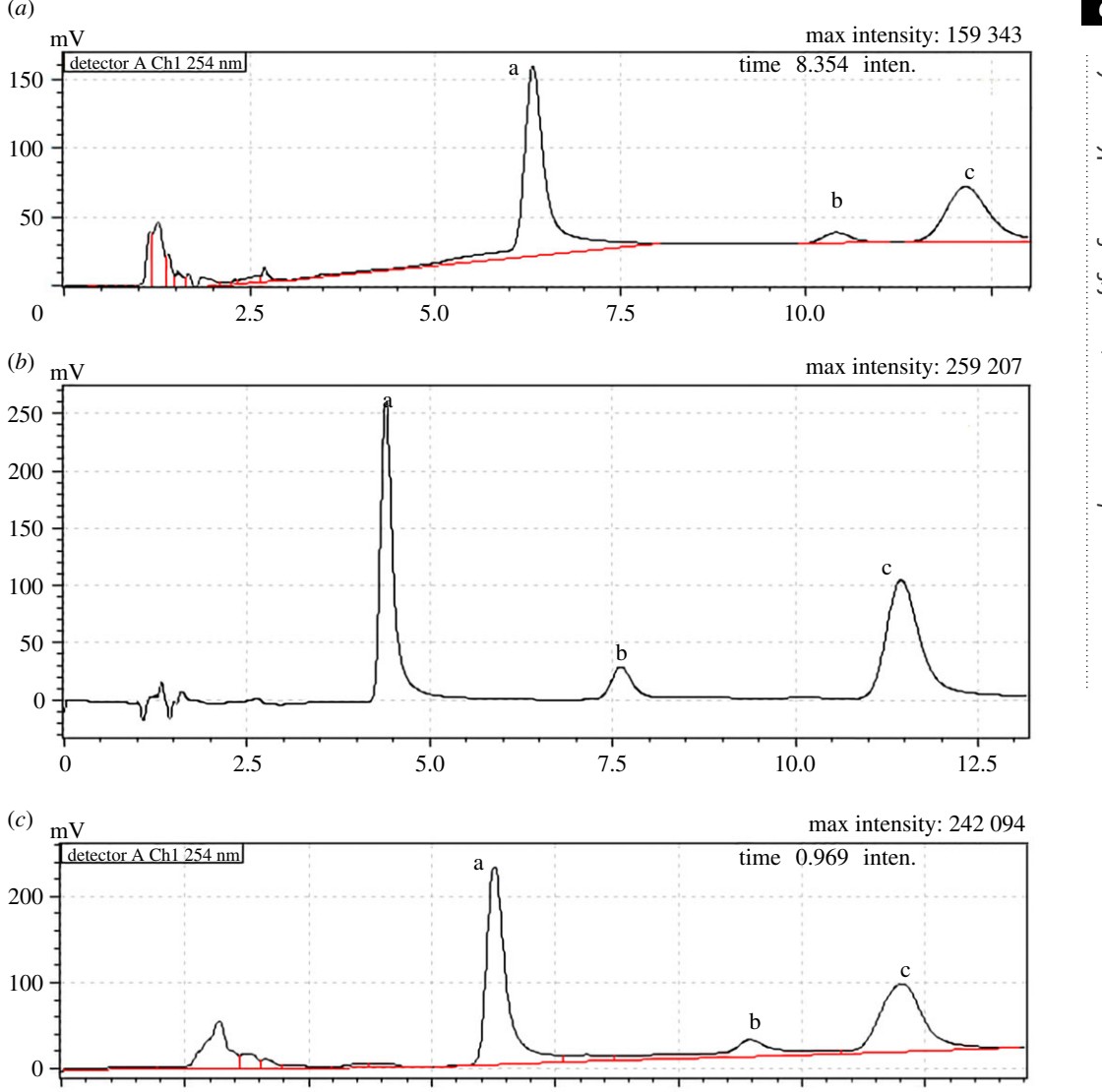

**Figure 1.** Representative chromatograms for (a) 50.0 μg ml$^{-1}$ ARP, (b) 10.0 μg ml$^{-1}$ PRM and (c) 50.0 μg ml$^{-1}$ PRB using the optimized chromatographic conditions of: (a) MLC, (b) HMLC and (c) MELC.

2.34 and 1.8 for ARP, PRM and PRB, respectively). These results could be explained by comprehending the separation retention mechanism offered by each mobile phase. When applied, MLC resulted in relatively lower values for the three drugs (figure 2a and table 2), which could be attributed to the adsorption of the surfactant monomers on the stationary phase surface, loading it with intense charges [25], which results in steric and electrostatic influences that exclude anionic micelles, hindering solute from accessing them [25], consequently resulting in lower partitioning of the drugs to micelles than to the aqueous phase. The results obtained suggest that MLC cannot be considered a suitable analysis tool for log P prediction, where the presence of the aqueous phase solely does not imitate the *in vivo* biological conditions, which is in accordance with an early study dealing with major aspects in micellar liquid chromatography (MLC) [26]. On the other hand, this report [26] emphasized the fact that the introduction of an organic solvent to MLC will significantly improve the obtained log $P_{m/w}$ values, where such mobile phases will mimic the partitioning of pharmaceuticals in the biological environment. Consequently, the introduction of *n*-octane and/or acetonitrile to the previously optimized MLC resulted in MELC and HMLC, respectively. When acetonitrile is introduced in HMLC, it enhances the separation efficiency by sweeping the SDS monomers from the surface of the stationary phase [25], allowing binding of solutes with the ionic micelles, and reducing the retention times of the analytes [25]. This mechanism could explain the comparable log $P_{m/w}$ values obtained for the studied pharmaceuticals with their reported log P values (figure 2b and table 2). On

royalsocietypublishing.org/journal/rsos　　R. Soc. Open Sci. **8**: 203371

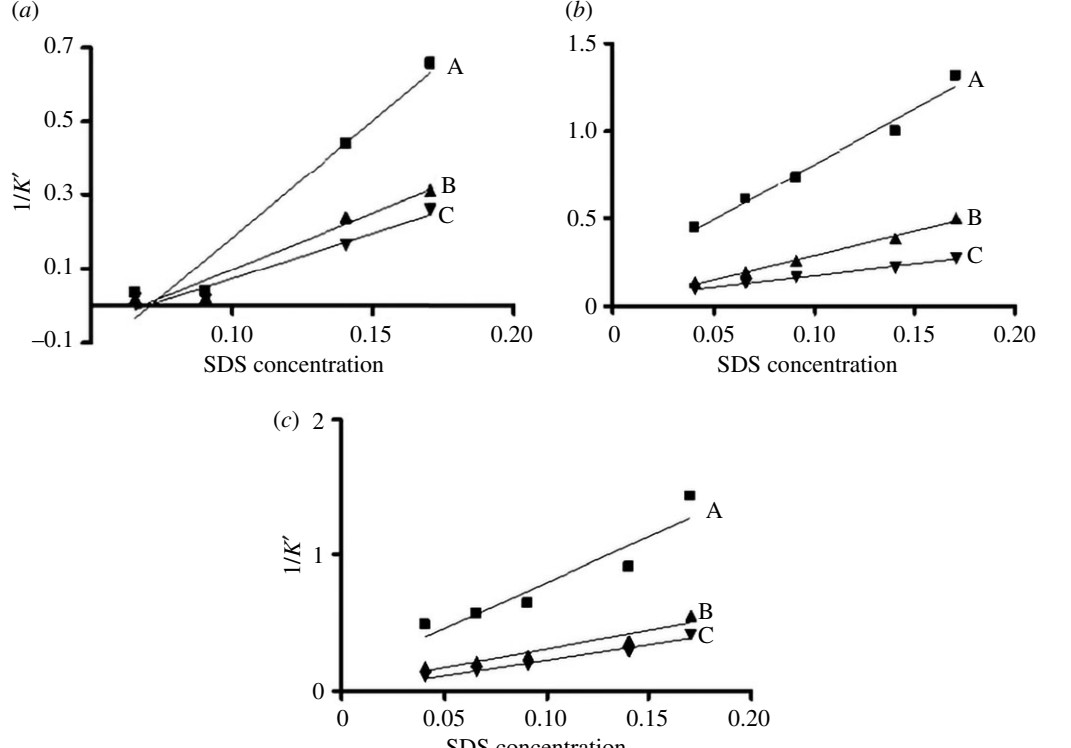

**Figure 2.** (a) Experimental determination of log $P_{m/w}$ values applying MLC (A: 50.0 µg ml$^{-1}$ ARP; B: 10.0 µg ml$^{-1}$ PRM and C: 50.0 µg ml$^{-1}$ PRB). (b) Experimental determination of log $P_{m/w}$ values applying HMLC (A: 50.0 µg ml$^{-1}$ ARP; B: 10.0 µg ml$^{-1}$ PRM and C: 50.0 µg ml$^{-1}$ PRB). (c) Experimental determination of log $P_{m/w}$ values applying MELC (A: 50.0 µg ml$^{-1}$ ARP; B: 10.0 µg ml$^{-1}$ PRM and C: 50.0 µg ml$^{-1}$ PRB).

**Table 2.** Comparison of experimental log $P_{m/w}$ values of the studied drugs using the three optimized micelle-based mobile phases with the reported values

| studied drug | experimental log $P_{m/w}$ | | | linear regression equations | | | reported log $P$ [23] |
|---|---|---|---|---|---|---|---|
| | MLC | HMLC | MELC | MLC | HMLC | MELC | |
| ARP | 2.182 | 4.729 | 2.275 | $Y = 2.182X - 0.969$ ($R^2 = 0.9555$) | $Y = 4.729X - 0.0027$ ($R^2 = 0.9813$) | $Y = 2.275X - 0.372$ ($R^2 = 0.8854$) | 4.6 |
| PRM | 0.9385 | 2.323 | 1.316 | $Y = 0.9385X - 0.47$ ($R^2 = 0.9507$) | $Y = 2.323X - 0.035$ ($R^2 = 0.9927$) | $Y = 1.316X - 0.122$ ($R^2 = 0.9261$) | 2.34 |
| PRB | 0.7334 | 1.648 | 1.491 | $Y = 0.7334X - 0.384$ ($R^2 = 0.9497$) | $Y = 1.648X + 0.023$ ($R^2 = 0.9948$) | $Y = 1.491X - 0.09$ ($R^2 = 0.9656$) | 1.8 |

the other hand, when MELC was used, the experimentally resultant log $P_{m/w}$ values were lower than expected (figure 2c and table 2); this outcome can be interpreted by understanding the potential retention behaviour in MELC. In MELC, the nano-diameter of the oil droplets (less than 10 nm) yields a remarkable interfacial area accompanied with enhanced interactions [27]. The droplets are included within the micelles in the solution, and acquire the same negative charge carried by the surfactant heads [27]. Lipophilic compounds like the investigated pharmaceuticals will penetrate and dissolve into micelle's core which is now more hydrophobic, through the interaction of oil droplets with non-polar tail of micelles. It is well established that hydrophobic drugs analysed by MELC undergo enhanced mass transfer from the lipophilic core of the micelles to the stationary phase, resulting in rapid elution of drugs [27]. Consequently, ARP, PRM and PRB are postulated to reside within the

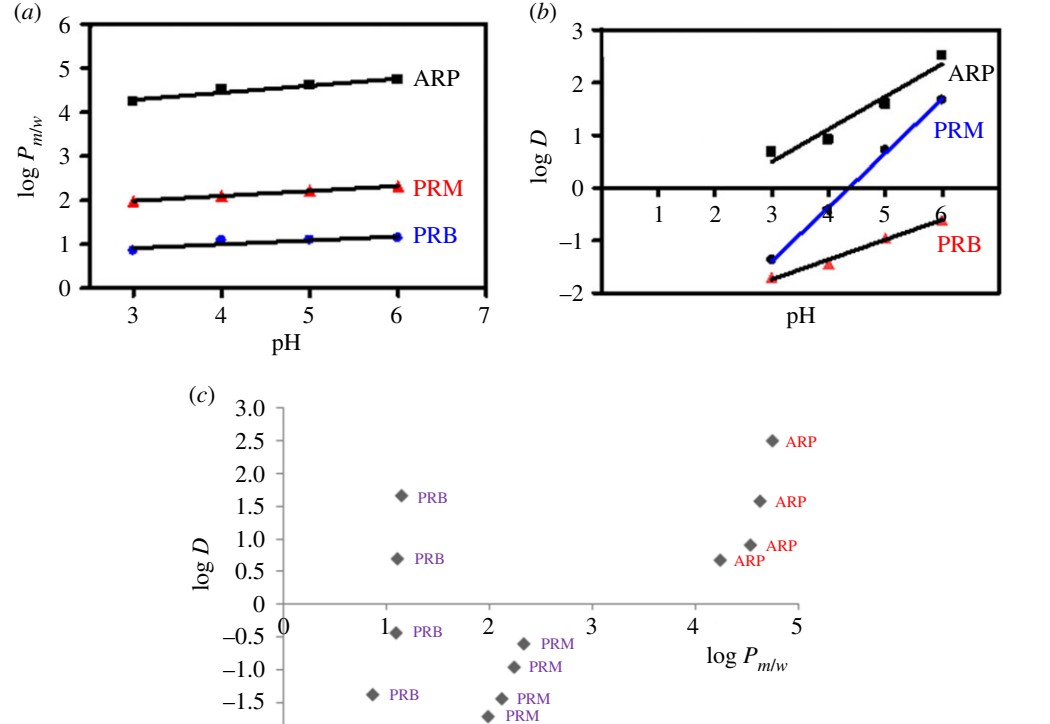

**Figure 3.** (*a*) Effect of pH of HMLC on log $P_{m/w}$ values of the studied drugs (ARP (50.0 µg ml$^{-1}$); PRM (10.0 µg ml$^{-1}$) and PRB (50.0 µg ml$^{-1}$)). (*b*) Effect of pH on the calculated log $D$ values of the studied drugs. (*c*) Correlation between experimental log $P_{m/w}$ obtained by HMLC and calculated log $D$ values of the studied drugs.

hydrophobic core for a limited time, hence their experimental log $P_{m/w}$ values are lower than that in HMLC (table 2).

As a conclusion, HMLC proved to be the most suitable micellar liquid chromatographic analysis tool for the experimental determination of log $P_{m/w}$ values of the studied drugs.

## 3.3. Effect of pH of HMLC on experimental log $P_{m/w}$ values of the studied drugs

One of the major factors that affect the experimentally determined log $P_{m/w}$ values is the pH of the mobile phase. As could be revealed from table 3 and figure 3*a*, increasing mobile phase pH (keeping column temperature during studying this parameter at 40°C) results in a subsequent increase in log $P_{m/w}$ for the three drugs. These results are expected, where the positive charges carried by the three analytes decrease by increasing pH, causing a subsequent increase in the analytes lipophilicities, which is expressed as an elevation in their log $P_{m/w}$ values.

The increase in log $P_{m/w}$ values of the three analytes with pH followed a linear pattern (figure 3*a*) over the applied pH range (3–6), where the regression equations of ARP, PRM and PRB were: log $P_{m/w}$ = 3.807 + 0.159 pH, log $P_{m/w}$ = 1.651 + 0.113 pH and log $P_{m/w}$ = 0.655 + 0.088 pH, with the corresponding $R^2$ values of 0.9239, 0.9965 and 0.9765.

The calculated log $D$ values (table 3) are considered an indication of the experimentally determined log $P_{m/w}$ of the three studied pharmaceuticals in spite of the significant variation in their absolute values, since the linear increase in log $D$ values for the drugs under study with subsequent increase in pH predicts the behaviour of log $P_{m/w}$ towards pH. The linear correlation of log $D$ with pH could be demonstrated in figure 3*b*, with the following regression equations for ARP, PRM and PRB: log $D$ = −1.365 + 0.62 pH, log $D$ = −2.878 + 0.379 pH and log $D$ = −4.465 + 1.025 pH. The corresponding $R^2$ values are 0.9392, 0.9878 and 0.9988. Eventually, the correlation between the calculated log $D$ and the practical log $P_{m/w}$ of the three analytes could be manifested in figure 3*c*.

It is worth mentioning that at pH 6—which was selected as the optimum value for separation—a significant accordance between the experimental, calculated and reported log *p*-values is observed (table 4), which emphasizes the suitability of the optimized HMLC as an analytical tool for practical determination of log $P$ of the studied compounds.

**Table 3.** Effect of HMLC pH (at 40°C) and column temperature (at pH 6) on the experimental log $P_{m/w}$ or/and calculated log $D$ values of the studied drugs.

| | pH 3 | | pH 4 | | pH 5 | | pH 6 | |
|---|---|---|---|---|---|---|---|---|
| studied drug | log $P$ | log $D$ | log $P$ | log $D$ | log $P$ | log $D$ | log $P$ | log $D$ |
| ARI | 4.229 | 0.68 | 4.522 | 0.91 | 4.615 | 1.59 | 4.729 | 2.52 |
| PRM | 1.985 | −1.70 | 2.109 | −1.44 | 2.228 | −0.95 | 2.323 | −0.60 |
| PRB | 0.8598 | −1.37 | 1.089 | −0.42 | 1.104 | 0.71 | 1.648 | 1.67 |
| | experimental log $P_{m/w}$ | | | | | | | |
| column temperature (°C) | ARI | | PRM | | | PRB | | |
| 25 | 3.12 | | 1.1714 | | | 0.976 | | |
| 30 | 3.66 | | 2.197 | | | 0.9707 | | |
| 35 | 4.46 | | 2.228 | | | 1.208 | | |
| 40 | 4.729 | | 2.323 | | | 1.648 | | |

**Table 4.** Comparison of the experimental, calculated and reported log $P$ of the studied drugs at pH 6 applying HMLC.

| studied drug | experimental log $P$ | calculated log $P$ | reported log $P$ [23] |
|---|---|---|---|
| ARI | 4.729 | 3.763 ± 0.520 | 4.6 |
| PRM | 2.323 | 2.347 ± 0.397 | 2.34 |
| PRB | 1.648 | 2.433 ± 0.517 | 1.8 |

## 3.4. Effect of column temperature on experimental log $P_{m/w}$ values of the studied drugs

Another major factor that influences the obtained log $P_{m/w}$ values is the column temperature; keeping in mind that pH is kept constant. Table 3 abridges the practical log $P_{m/w}$ values of ARP, PRM and PRB upon changing the column temperature over the range of 25–40°C, where HMLC mobile phase pH was kept at 6. By increasing the temperature, the obtained log $P_{m/w}$ values of all the drugs increase. This phenomenon is expected, as enhanced partitioning of solutes to micelles by elevating the temperature is a well-established concept [12]. It is worth mentioning that raising the temperature settings above 40°C is not advisable, as micelles have a potential solubilizing ability of silanol groups of the stationary phase [26].

## 3.5. Method validation

The developed HMLC method was subjected to the validation procedure assessed by the United States Pharmacopeia (USP) [28], to verify its utility for its intended purpose of practical log $P_{m/w}$ determination. Linearity ranges were found to be (1.0–200.0, 0.5–100.0 and 1.0–250.0 µg ml$^{-1}$) for ARP, PRM and PRB, respectively, with limits of detection LOD of 0.4, 0.3 and 0.5 µg ml$^{-1}$, respectively, and the corresponding quantification limits LOQ of 0.7, 0.4 and 0.8 µg ml$^{-1}$. The LOD and LOQ were determined experimentally by calculating the concentrations yielding peaks to noise ratios of 3 : 1 (LOD) or 10 : 1 (LOQ) as stated by the USP [28]. The proposed and reference methods [29–31] were applied to determine the concentration of each drug either alone or in laboratory-prepared mixtures. By recognition of the high values of the percentage recoveries, and low values of $t$ and $F$ parameters [32], the accuracy of the proposed method could be easily proved (table 5).

Moreover, inter- and intra-day precision using three different concentrations of each drug on 3 days or on the same day, respectively, were investigated. The proposed HMLC method was found to be precise as could be indicated from the low values of the standard deviation (table 5).

**Table 5.** Accuracy and precision data for the proposed HMLC method.

| studied drug | parameter | % found | reference methods, % found [29–31] |
|---|---|---|---|
| *application of HMLC for the determination of the drugs in pure form* | | | |
| ARP | mean ± s.d. | 100.49 ± 1.32 | 100.74 ± 1.15 |
| | t | 0.07 (2.015)[a] | |
| | F | 1.32 (19.3) | |
| PRM | mean ± s.d. | 99.92 ± 1.05 | 100.62 ± 1.27 |
| | t | 0.19 (2.015) | |
| | F | 1.46 (19.3) | |
| PRB | mean ± s.d. | 100.09 ± 1.39 | 99.57 ± 1.75 |
| | t | 0.21 (2.015) | |
| | F | 1.59 (19.3) | |
| *application of HMLC for the determination of the studied drugs in laboratory-prepared mixtures* | | | |
| ARP | mean ± s.d. | 100.09 ± 1.63 | 100.14 ± 1.44 |
| | t | 0.28 (2.776) | |
| | F | 1.28 (19.25) | |
| PRM | mean ± s.d. | 100.31 ± 1.17 | 100.49 ± 1.21 |
| | t | 0.82 (2.776) | |
| | F | 1.51 (19.25) | |
| PRB | mean ± s.d. | 99.67 ± 0.46 | 100.51 ± 1.14 |
| | t | 0.62 (2.776) | |
| | F | 6.14 (19.25) | |

| studied drug | parameter | inter-day precision | intra-day precision |
|---|---|---|---|
| *precision data of the proposed HMLC method* | | | |
| ARP | mean ± s.d. | 99.98 ± 0.75 | 100.24 ± 0.74 |
| PRM | | 99.98 ± 0.43 | 100.36 ± 1.41 |
| PRB | | 100.26 ± 0.61 | 99.76 ± 0.92 |

[a]Figures between parentheses are the tabulated *t* and *F* values at *P* = 0.05 [32].

The proposed method also demonstrated a significant robustness, as all the studied concentrations of SDS (0.05–0.18) succeeded to elute the three drugs despite the decrease in their retention times by elevation in SDS concentrations. Besides, the retention times of all drugs were slightly affected by the change in pH (3–6). Moreover, acetonitrile ratios (from 7 to 15%) did not affect $t_r$ or the sensitivity of any of the three drugs.

# 4. Conclusion

A detailed investigation was carried out to study the suitability of HMLC for the practical determination of log *P* of three selected PDA used to attain the reward circuit. The obtained results suggest that HMLC is preferential to MLC and MELC in deducing this significant physico-chemical parameter, where the obtained log $P_{m/w}$ values were in accordance with the reported and the calculated log *P* values, unlike the other mobile phases. Explanation of the obtained results was introduced based on documented retention mechanism of each mobile phase. Besides, the effect of pH and column temperature of the developed HMLC mobile phase on log $P_{m/w}$ was also investigated. Eventually, full validation of HMLC was carried out to ensure its suitability for the intended purpose.

Data accessibility. Data representing selected chromatograms are available at the Dryad Digital Repository: https://doi.org/10.5061/dryad.q2bvq83hv. Retention behavior of analytes applying three micellar mobile phases available at: https://doi.org/10.5061/dryad.qv9s4mwd8.

Authors' contributions. M.E.K.W.: designed the study, interpreted the results and wrote the draft of the manuscript. D.E.S.: shared in data analysis, shared in interpretation of the results and revised the manuscript. D.E.W.: shared in statistical analysis of the data, shared in interpretation of the results and revised the manuscript.

Competing interests. We declare we have no competing interests.

Funding. There is no funding—for any of the authors—to report for this article.

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
