## [Peer Review File · Royal Society Open Science]

Review History

RSOS-202371.R0 (Original submission)

Review form: Reviewer 1

Is the manuscript scientifically sound in its present form?

Yes

Are the interpretations and conclusions justified by the results?

Yes

Is the language acceptable?

Yes

Do you have any ethical concerns with this paper?

No

Have you any concerns about statistical analyses in this paper?

No

Recommendation?

Accept as is

Comments to the Author(s)

I think this is a well organized and well written manuscript and can be accepted as is. I think the discussion is convincing and conclusions are supported by enough data, tables and figures. In addition, it is well written with minimal typo or language errors.

Review form: Reviewer 2**Is the manuscript scientifically sound in its present form?**

No

Are the interpretations and conclusions justified by the results?

Yes

Is the language acceptable?

No

Do you have any ethical concerns with this paper?

No

Have you any concerns about statistical analyses in this paper?

No

Recommendation?

Major revision is needed (please make suggestions in comments)

Comments to the Author(s)

It is an interesting work regarding the calculation of log P_{m/w} values for some partial dopamine agonists (PDA) namely aripiprazole, pramipexole, and piribedil by micellar liquid chromatography (MLC), hybrid micellar liquid chromatography (HMLC), and micro emulsion liquid chromatography (MELC). The research theme and the experimental work presented in this manuscript are interesting and to me this work deserves publication after major revision.

Comments

The overall quality of the paper regarding readability and language is poor. The paper reads as an early draft with deficiencies in organization which are difficult to read. The authors should make significant effort to further revise the manuscript in accordance with Royal Society Open Science guidelines.

Page 2, line 6, an explanation is needed for the acronyms CMC of SDS .

Page 7, line 26 Please mention the main reasons (characteristics of stationary phase) that lead you to the choice of ProntoSil Kromaplus C18 analytical column for this research.

Page 7, line 36 the reason of setting micellar mobile phase to pH 6, should be mentioned.

Page 8, Line 40, Log P_{m/w}

Page 8 lines 45 to 50: The authors mention that "Actual concentrations of SDS for HMLC were plotted versus the inverse of the capacity factor for each of ARP, PRM, and PRB when the pH of the mobile phase was modified over the range of (3-6), keeping the column temperature at 40°C".

I guess in y-axis is SDS concentration and x-axis inverse of capacity factor... is this a 3D diagram including pH variation or several different diagrams, please rephrase this sentence.

Page 9 line 3 the program used for to calculate dissociation constant (log D) should be mentioned.

Page 10 line 12, 0.05 M succeeded to elute the three drugs but with long unacceptable retention times accompanied with band broadening of both PRM and PRB... You could attribute peak tailing to the interaction of the residual silanols with these basic compounds. The addition of triethylamine is one way to reduce these silanophilic interactions.

Page 12, line 35, please mention the experimental log P o/w values

Page 12: lines 33-52 this paragraph about mass transfer should be re-written. At pH 6 these compounds are positively charged therefore LogD is seriously reduced

Page 12, lines 22-23, Since the main reason of this research was to study their lipophilicity, I don't understand the reason why it was so important to separate these three drugs?

Figures 2 to 4, reduce the thickness of trendline, correlation coefficients should be reported
Figures and tables should be reduced.

Decision letter (RSOS-202371.R0)

Dear Dr Wahba:

Title: suitability of hybrid micelle liquid chromatography for estimating lipophilicity of partial dopamine agonists used to attain the reward circuit

Manuscript ID: RSOS-202371

The editor assigned to your manuscript has now received comments from reviewers. I apologise that this has taken longer than usual. We would like you to revise your paper in accordance with the referee and Subject Editor suggestions which can be found below (not including confidential reports to the Editor). Please note this decision does not guarantee eventual acceptance.

Please submit your revised paper before 14-Apr-2021. Please note that the revision deadline will expire at 00.00am on this date. If we do not hear from you within this time then it will be assumed that the paper has been withdrawn. In exceptional circumstances, extensions may be possible if agreed with the Editorial Office in advance. We do not allow multiple rounds of revision so we urge you to make every effort to fully address all of the comments at this stage. If deemed necessary by the Editors, your manuscript will be sent back to one or more of the original reviewers for assessment. If the original reviewers are not available we may invite new reviewers.

RSC Associate Editor:
Comments to the Author:
(There are no comments.)

RSC Associate Editor:
Comments to the Author:
(There are no comments.)

Reviewers' Comments to Author:
Reviewer: 1

Comments to the Author(s)

I think this is a well organized and well written manuscript and can be accepted as is. I think the discussion is convincing and conclusions are supported by enough data, tables and figures. In addition, it is well written with minimal typo or language errors.

Reviewer: 2

Comments to the Author(s)

It is an interesting work regarding the calculation of log P_{m/w} values for some partial dopamine agonists (PDA) namely aripiprazole, pramipexole, and piribedil by micellar liquid chromatography (MLC), hybrid micellar liquid chromatography (HMLC), and micro emulsion liquid chromatography (MELC). The research theme and the experimental work presented in this manuscript are interesting and to me this work deserves publication after major revision.

Comments

The overall quality of the paper regarding readability and language is poor. The paper reads as an early draft with deficiencies in organization which are difficult to read. The authors should make significant effort to further revise the manuscript in accordance with Royal Society Open Science guidelines.

Page 2, line 6, an explanation is needed for the acronyms CMC of SDS .

Page 7, line 26 Please mention the main reasons (characteristics of stationary phase) that lead you to the choice of ProntoSil Kromaplus C18 analytical column for this research.

Page 7, line 36 the reason of setting micellar mobile phase to pH 6, should be mentioned.

Page 8, Line 40, Log P_{m/w}

Page 8 lines 45 to 50: The authors mention that "Actual concentrations of SDS for HMLC were plotted versus the inverse of the capacity factor for each of ARP, PRM, and PRB when the pH of the mobile phase was modified over the range of (3-6), keeping the column temperature at 40°C". I guess in y-axis is SDS concentration and x-axis inverse of capacity factor... is this a 3D diagram including pH variation or several different diagrams, please rephrase this sentence.

Page 9 line 3 the program used for to calculate dissociation constant (log D) should be mentioned.

Page 10 line 12, 0.05 M succeeded to elute the three drugs but with long unacceptable retention times accompanied with band broadening of both PRM and PRB... You could attribute peak tailing to the interaction of the residual silanols with these basic compounds. The addition of triethylamine is one way to reduce these silanophilic interactions.

Page 12, line 35, please mention the experimental log P_{o/w} values

Page 12: lines 33-52 this paragraph about mass transfer should be re-written. At pH 6 these compounds are positively charged therefore LogD is seriously reduced

Page 12, lines 22-23, Since the main reason of this research was to study their lipophilicity, I don't understand the reason why it was so important to separate these three drugs?

Figures 2 to 4, reduce the thickness of trendline, correlation coefficients should be reported
Figures and tables should be reduced.

Author's Response to Decision Letter for (RSOS-202371.R0)

See Appendix A.

Decision letter (RSOS-202371.R1)

Dear Dr Wahba:

Title: suitability of hybrid micelle liquid chromatography for estimating lipophilicity of partial dopamine agonists used to attain the reward circuit

Manuscript ID: RSOS-202371.R1

It is a pleasure to accept your manuscript in its current form for publication in Royal Society Open Science. The chemistry content of Royal Society Open Science is published in collaboration with the Royal Society of Chemistry.

Please see the Royal Society Publishing guidance on how you may share your accepted author manuscript at <https://royalsociety.org/journals/ethics-policies/media-embargo/>. After publication, some additional ways to effectively promote your article can also be found here

<https://royalsociety.org/blog/2020/07/promoting-your-latest-paper-and-tracking-your-results/>.

RSC Associate Editor
Comments to the Author:
(There are no comments.)

Reviewer(s)' Comments to Author:

Appendix A

From the Office of
Dr. Mary Wahba
Pharm. Anal. Chem. Dept.
Faculty of Pharmacy
University of Mansoura
Mansoura 35516
Egypt.
Fax: 00(20)502247496
E-mail: marywahba5@gmail.com

March, 29th, 2021

Dear Prof.:

Enclosed, please find the point by point respond to reviewers' comments on the manuscript entitled: **Studying the suitability of hybrid micelle liquid chromatography for estimating the lipophilicity of some partial dopamine agonists used to attain the reward circuit**

Thank you for your kind cooperation and I am looking forward to hearing from you soon.

Sincerely Yours,

Dr. Mary Wahba

Reviewer#1

Comments to the Author(s)

I think this is a well organized and well written manuscript and can be accepted as is. I think the discussion is convincing and conclusions are supported by enough data, tables and figures. In addition, it is well written with minimal typo or language errors.

Reviewer#2

Comments to the Author(s)

It is an interesting work regarding the calculation of log P_{m/w} values for some partial dopamine agonists (PDA) namely aripiprazole, pramipexole, and piribedil by micellar liquid chromatography (MLC), hybrid micellar liquid chromatography (HMLC), and micro emulsion liquid chromatography (MELC). The research theme and the experimental work presented in this manuscript are interesting and to me this work deserves publication after major revision.

Comments

Page 2, line 6, an explanation is needed for the acronyms CMC of SDS.

An explanation for the acronyms CMC of SDS has been given.

Page 7, line 26 Please mention the main reasons (characteristics of stationary phase) that lead you to the choice of ProntoSil Kromaplus C18 analytical column for this research.

The main reasons leading to the choice of ProntoSil Kromaplus C18 analytical column for this research were mentioned

Page 7, line 36 the reason of setting micellar mobile phase to pH 6, should be mentioned.

Explanation of setting micellar mobile phase to pH 6 has been mentioned in details under section (3.1.Separation parameters optimization), page 9 last paragraph and

page 10 first paragraph.

Page 8, Line 40, Log P_{m/w}

Correction was made

Page 8 lines 45 to 50: The authors mention that "Actual concentrations of SDS for

HMLC were plotted versus the inverse of the capacity factor for each of ARP, PRM, and PRB when the pH of the mobile phase was modified over the range of (3-6), keeping the column temperature at 40°C". I guess in y-axis is SDS concentration and x-axis inverse of capacity factor... is this a 3D diagram including pH variation or several different diagrams, please rephrase this sentence.

The paragraph was re-phrased to be clear to the readers. This figure is not a 3D diagram, it represent different diagrams. (A: ARP; B: PRM and C: PRB). Five sets of HMLC mobile phases were prepared; each set has a constant SDS concentration (either 0.05 M,0.075 M,0.1 M,0.15 M or 0.18 M), and each set is prepared with different pH values (ranging from 3-6). Actual concentrations of SDS in the five sets (X axis) were plotted versus the inverse of the capacity factor for each of ARP, PRM, and PRB (Y axis), where analysis was performed keeping the column temperature at 40°C.

Page 9 line 3 the program used for to calculate dissociation constant (log D) should be mentioned.

The program used for to calculate dissociation constant (log D) has been mentioned under section (2.1. Instrumentation)

Page 10 line 12, 0.05 M succeeded to elute the three drugs but with long unacceptable retention times accompanied with band broadening of both PRM and PRB... You could attribute peak tailing to the interaction of the residual silanols with these basic compounds. The addition of triethylamine is one way to reduce these silanophilic interactions.

An explanation of peaks tailing was presented focusing on the role of triethyl amine regarding this aspect

Page 12, line 35, please mention the experimental log P o/w values

The experimental log P o/w values has been mentioned as requested

Page 12: lines 33-52 this paragraph about mass transfer should be re-written.

The paragraph concerned with mass transfer was re written to be clear to the readers.

Page 12, lines 22-23, Since the main reason of this research was to study their lipophilicity, I don't understand the reason why it was so important to separate these three drugs?

The accuracy of the experimentally determined LogP values -which reflect the lipophilicity of the studied compounds- depends mainly on applying the HMLC optimized separation parameters. Hence, separation of the three drugs was essential to conclude the optimum condition required to calculate their Log P values. Besides, the developed separation conditions could be useful for other analysts who might need to apply it in a different application. In addition, to guarantee that applicability of the proposed method for its intended use, it should be fully validated, which in turn requires optimization of the separation conditions.

Figures 2 to 4, reduce the thickness of trendline, correlation coefficients should be reported

Thickness of trend line was reduced in the three figures. Regression equations together with correlation coefficients were included in Table 2.

Figures and tables should be reduced.

Reduction in the number of figures and tables was carried out as follows: Figures 2, 3 and 4 were merged into one figure. Figures 5, 6, and 7 were merged into one figure. The data in table 5 were merged into table 3. Data in tables 6, 7, and 8 were merged in a single table. Accordingly, re numbering of figures and tables was carried out all over the manuscript.

All changes in the manuscript are marked with red bold font.